# Persistence of Antibiotic-Resistant *Escherichia coli* Strains Belonging to the B2 Phylogroup in Municipal Wastewater under Aerobic Conditions

**DOI:** 10.3390/antibiotics11020202

**Published:** 2022-02-04

**Authors:** Hui Xie, Yoshitoshi Ogura, Yoshihiro Suzuki

**Affiliations:** 1Department of Civil and Environmental Engineering, Faculty of Engineering, University of Miyazaki, Miyazaki 889-2192, Japan; z370702@student.miyazaki-u.ac.jp; 2Division of Microbiology, Department of Infectious Medicine, Kurume University School of Medicine, Fukuoka 830-0011, Japan; y_ogura@med.kurume-u.ac.jp

**Keywords:** *Escherichia coli*, antibiotic resistant, phylogroups, municipal wastewater

## Abstract

*Escherichia coli* is classified into four major phylogenetic groups (A, B1, B2, and D) that are associated with antibiotic resistance genes. Although antibiotic-resistant *E. coli* is commonly detected in municipal wastewater, little is known about the relationship between the phylogenetic groups and antibiotic-resistant *E. coli* in wastewater. In this study, the survival of *E. coli* in wastewater and the changes to the relationships between each phylogroup and the antibiotic-resistant profiles of *E. coli* isolates from wastewater were investigated under aerobic conditions for 14 days. The isolates were classified into the phylogroups A, B1, B2, and D or others by multiplex PCR. In addition, the susceptibility of the isolates to 11 antibiotics was assessed with the minimum inhibitory concentration (MIC) assay. While *E. coli* counts decreased in the wastewater with time under aerobic conditions, the prevalence of phylogroup B2 had increased to 73% on day 14. Furthermore, the MIC assay revealed that the abundance of antibiotic-resistant *E. coli* also increased on day 14. After batch-mixing the experiments under aerobic conditions, the surviving antibiotic-resistant *E. coli* included mainly multidrug-resistant and beta-lactamase-producing isolates belonging to phylogroup B2. These results suggest that the phylogroup B2 isolates that have acquired antibiotic resistance had a high survivability in the treated wastewater.

## 1. Introduction

The emergence and spread of antimicrobial resistance poses serious threats to the health of animals and humans as antibiotic-resistant bacteria account for about 700,000 deaths annually worldwide [1] Despite the absence of reliable data on the number of deaths caused by antibiotic-resistant bacteria in Japan, 8000 deaths were confirmed in 2017 due to two representative types of antibiotic-resistant bacteria: methicillin-resistant *Staphylococcus aureus* and fluoroquinolone-resistant *Escherichia coli* [2]. The number of deaths due to antibiotic-resistant bacteria is predicted to increase to 10 million globally by 2050, confirming that antibiotic resistance poses a significant threat to human health and warranting further studies to prevent and reduce the risk of infection with antibiotic-resistant bacteria. In consideration of these serious circumstances, in 2017 the World Health Organization listed *E. coli*, *Shigella*, and *Salmonella* in the *Enterobacteriaceae* family as antibiotic-resistant bacteria that pose serious threats to human health and released survey data to warn the global population of the potential severity.

Members of the *Enterobacteriaceae* family include pathogens associated with infections of the gastrointestinal and urinary tracts [3,4]. *E. coli*, which is considered the most important bacterium of the intestinal flora of humans and animals, can survive in treated wastewater and can spread through wastewater treatment plants to rivers and other aquatic environments. *E. coli* strains that are resistant to innovative effective antimicrobial agents, such as third- and fourth-generation cephalosporins and carbapenems, have been detected in natural environments [5,6,7]. Notably, the percentage of *E. coli* isolates resistant to third-generation cephalosporins increased from 70% to 83% in India between 2008 and 2013 [8]. In Europe, there has been a steady increase in the detection of strains producing extended-spectrum beta-lactamases (ESBLs) that are resistant to third-generation cephalosporins [9].

Multidrug-resistant and ESBL-producing *E. coli* have been detected at relatively high concentrations in both treated and untreated wastewater samples [10,11]. Furthermore, antibiotic-resistant *E. coli* has been reported in rivers and other aquatic environments [12,13,14]. Since *E. coli* can survive in wastewater treatment systems, it is critical to track the source of antibiotic-resistant strains in aquatic environments, especially multidrug-resistant and ESBL-producing *E. coli*, which pose a grave threat to public health [15,16,17,18].

Based on gene structure and sequence data, *E. coli* is commonly classified into four major phylogenetic groups (A, B1, B2, and D), which exhibit differences in ecological specificity [19]. Since the host and environmental factors impact the diversity and abundance of *E. coli*, the source of the host and the pathogenicity of *E. coli* strains can be distinguished by gene sequencing technology [20,21,22,23]. In general, *E. coli* strains of phylogroups A and B1 are most likely to inhabit the intestinal tracts of humans and animals. Symbiotic strains isolated from humans mainly belong to phylogroup A [21], while most strains isolated from animals belong to phylogroup B1 [24]; strains isolated from wastewater systems mainly belong to phylogroups B2 and D [25]. However, there is little information about the relationship between the phylogroups and antibiotic-resistant *E. coli* in municipal wastewater [26], especially for the processing of wastewater. Hence, the aim of the present study was to investigate changes to the relationship between each phylogroup and antibiotic-resistant profiles of *E. coli* isolates from municipal wastewater under aerobic mixing conditions for 14 days via batch experiments.

## 2. Results

### 2.1. Changes to TOC Concentrations and Number of E. coli Colonies under Aerobic Conditions

Changes to the TOC concentrations and the number of *E. coli* colonies in wastewater under aerobic conditions are shown in Appendix A. The other parameters of water quality and the number of bacteria are presented in Appendix A. The TOC concentration gradually decreased after 14 days under aerobic conditions. The TOC concentrations of wastewater samples collected from plants A and B decreased from 55.4 to 11.0 mg/L and from 53.5 to 12.7 mg/L after 14 days, respectively, while turbidity decreased from 102.4 to 59.1 and from 114.8 to 53.2. These findings confirmed the decomposition of organic matter in wastewater under aerobic conditions for 14 days.

The number of *E. coli* CFUs in the wastewater samples collected from plants A and B decreased from 6.9 × 10^6^ and 7.3 × 10^6^ CFU/100 mL on day 0 to 2.6 × 10^4^ and 9.3 × 10^4^ CFU/100 mL on day 14, respectively, under aerobic conditions. Under aerobic mixing conditions, the removal efficiency of *E. coli* from the samples collected from plants A and B was 99.6% and 98.7%, respectively. The decreases in *E. coli* content confirmed successful biological treatment of the wastewater under aerobic conditions.

### 2.2. Identification of E. coli Isolates

In the batch-mixing experiment, MALDI-TOF MS was used to identify *E*. *coli* in all 600 wastewater samples collected from plants A and B (days 0, 7, and 14). Of the 300 samples collected from plant A, 296 (98.7%) were positive for *E. coli*, three were positive for *Enterobacter cloacae,* and one was positive for *Citrobacter braakii*; meanwhile, all 300 samples collected from plant B were positive for *E. coli*. These results confirm that *E. coli*, but not the pseudo-positive strains, were prevalent in the wastewater collected from plants A and B.

### 2.3. Phylogroups of E. coli

Changes to the composition of the *E. coli* phylogroups occurred over time in the municipal wastewater samples collected from plants A and B under aerobic conditions (Figure 1). In the samples collected from plants A and B, 94.3% (279/296) and 82.3% (247/300) of isolates, respectively, were identified as the major phylogroups (A, B1, B2, and D). Before aerobic treatment, the most dominant phylogroup was B2 in the samples collected from both plants. Interestingly, after aerobic treatment, although the abundances of the major phylogroups other than B2 were considerably decreased in both plant samples, the abundance of phylogroup B2 was constant in the samples collected from plant B and had increased in those collected from plant A (Figure 1). These results indicate that the survival rate of the strains belonging to phylogroup B2 was much higher than that of the strains belonging to the other major phylogroups in the municipal wastewater samples under aerobic conditions.

### 2.4. Antibiotic Susceptibility of E. coli

All 596 *E. coli* isolates were tested for susceptibility to 11 antibiotics to assess changes to the prevalence of antibiotic-resistant *E. coli* in the wastewater samples collected from plants A and B over a 2-week period under aerobic conditions. Here, prevalence was defined as the percentage of isolates from each sample that was resistant to one or more antibiotics. Although the abundance of *E*. *coli* was relatively lower in the samples collected from plant A (Appendix A), the prevalence of antibiotic-resistant *E. coli* isolates increased from 14% (14/99) on day 0 to 44% (44/99) on day 14 (Figure 2, Appendix A). In contrast, the prevalence of antibiotic-resistant *E*. *coli* isolates in the samples collected from plant B increased from 69% (69/100) on day 0 to 79% (79/100) on day 7 and then decreased to 48% (48/100) on day 14. In the wastewater samples from plant B, the abundance of *E. albertii* had increased under aerobic conditions (Figure 1) by 16% between day 0 and day 14. Notably, 14% and 69% of the isolates in the wastewater samples collected from plants A and B, respectively, were resistant to at least one of the 11 antibiotics. In contrast, 79% and 48% of the isolates in the wastewater samples collected from plant B were resistant to at least one antibiotic on day 7 and 14, respectively. In a previous study, 44% of *E. coli* isolates in urban wastewater were resistant to at least one of the nine tested antibiotics [27].

The changes to the prevalence of isolates resistant to each tested antibiotic over the 14-day period are shown in Appendix A. The prevalence of isolates from plant A resistant to any of the 11 antibiotics had increased with time; especially, those resistant to AMP, CFZ, CIP, and TET, which had increased by more than 30% on day 14. The significant increase in the prevalence of antibiotic-resistant isolates was responsible for the overall increase in Figure 2. In contrast, the prevalence of isolates in the wastewater samples collected from plant B resistant to AMP, CFZ, and CTX was relatively high on day 0 and then significantly decreased on day 14. The rapid increase in the abundance of TET-resistant isolates on day 7 was responsible for the overall increase in prevalence in Figure 2. The *E*. *coli* isolates in two of the wastewater samples collected from plants A and B were susceptible to IMP.

The *E*. *coli* isolates in the wastewater samples collected from plants A and B were highly resistant to AMP, CFZ, CTX, TET, and CIP. However, there was no significant increase in the prevalence of resistant isolates in wastewater samples collected from plant B over the 14-day experimental period. According to the 2018 national action plan on antimicrobial resistance [28], the prevalence of *E*. *coli* isolates resistant to the penicillin antibacterial, AMP, and piperacillin had increased between 2011 and 2017 and remained greater than 40% for 6 consecutive years. In addition, approximately 20% of *E*. *coli* isolates are resistant to cephalosporins. The antibiotic susceptibility results of *E. coli* in this study were consistent with the trends described in the Nippon Antimicrobial Resistance One Health Report.

Information on multidrug-resistant bacteria is extremely important from a public health perspective. The prevalence of multidrug-resistant strains in the wastewater samples collected from plants A and B is shown in Figure 2 and Appendix A. By day 14, the prevalence of multidrug-resistant *E. coli* strains increased to 35% (35/99) in the plant A samples but decreased to 22% (22/100) in the plant B samples. *E. coli* strains resistant to five antibiotics were also detected in the wastewater samples collected from both plants A and B, but the prevalence was six-fold greater in samples from plant B compared with those from plant A. The prevalence of multidrug-resistant isolates in the wastewater samples collected from plant A had increased under aerobic conditions, while more than 10% of the isolates from plant B were resistant to five or more antibiotics. These findings suggest that the prevalence of antibiotic-resistant *E. coli* isolates differs among wastewater samples and is increased under aerobic conditions.

### 2.5. ESBL-Producing E. coli Isolates and Genotypes of E. coli

Testing for ESBL-producing genes in multidrug-resistant bacteria showed that 25 (8.4%) of 296 and 51 (17%) of 300 isolates in the wastewater samples collected from plants A and B, respectively, were ESBL-producing *E. coli*. The relationship between phylogroups and ESBL-producing *E. coli* was also evaluated based on the phylogroup analysis and ESBL-production profiles (Figure 3, Table 1). While *E. coli* counts gradually decreased with time under aerobic conditions, the prevalence of ESBL-producing *E. coli* accounted for 23% of the surviving *E. coli* isolates in the wastewater samples collected from plant A on day 14. The prevalence of ESBL-producing *E. coli* had increased in the wastewater samples collected from plant A, but not plant B, from day 0 to day 14. Most of the ESBL-producing isolates were classified as phylogroup B2, while one was classified as phylogroup B1. The CTX-M-9 type was dominant among the isolates in the wastewater samples collected from plant A. In contrast, the prevalence of ESBL-producing *E. coli* under aerobic conditions for 14 days did not increase in the samples from plant B, of which the SHV and CTX-M-9 types (29 and 22 isolates, respectively) were predominant, as determined by PCR analysis. All five phylogroups were identified in the samples collected from plant B and were almost evenly divided among phylogroups B2, B1, A, and D, and to a lesser extent in phylogroup F (24, 10, 7, 7, and 3 isolates, respectively). The phylogroups of isolates from plant B were more complex and diverse as compared with those from plant A. The survival rate of phylogroup B2 isolates carrying beta-lactamase genes was relatively high under aerobic conditions.

## 3. Discussion

### 3.1. Changes to the Composition of E. coli Phylogroups

In the samples collected from plants A and B 94.3% (279/296) and 82.3% (247/300) of isolates, respectively, were identified as the major phylogroups (A, B1, B2, and D) (Figure 1). Before aerobic treatment, the most dominant phylogroup was B2 in the samples collected from both plants. Interestingly, after aerobic treatment, although the abundances of the major phylogroups other than B2 were considerably decreased in both plant samples, the abundance of phylogroup B2 was constant in the samples collected from plant B and had increased compared with those from plant A (Figure 1). These results indicate that the survival rate of the strains belonging to phylogroup B2 was much higher than the strains belonging to the other major phylogroups in the municipal wastewater samples under aerobic conditions. Among various pathotypes of pathogenic *E. coli*, extraintestinal pathogenic *E. coli* (ExPEC) causes various human infections, including life-threatening sepsis and neonatal meningitis, thereby posing a significant public health concern [29]. ExPEC strains mostly belonged to phylogroup B2 [30,31]. In this study, the phylogroup B2 strains were the most predominant in the wastewater samples collected from both treatment plants in different areas within the same city and exhibited the greatest tolerance among all *E*. *coli* strains to the aerobic treatment of wastewater. From a public health point of view the survival of the phylogroup B2 strains in the treated wastewater poses serious public health concerns. *E. coli* phylogroups D and B2 are dominant in wastewater treatment plants in subtropical regions [25,32]. However, since the population structure of *E. coli* strains in wastewater is greatly affected by various factors such as temperature, treatment processes, and sampling regions further research in this field is required to support our findings. 

In this study, among the strains identified as *E. coli* by MALDI-TOF MS, 48 were actually *E. albertii* as determined by the phylotyping method developed by Clermont et al., [33]. Although *E. albertii* strains are often misidentified as *E. coli* by routine phenotyping methods and even MALDI-TOF MS [34], several studies have supported the classification of *E. albertii* as a distinct species in the genus *Escherichia* [35,36,37,38]. More importantly, *E. albertii* was recently recognized as an emerging human pathogen, which encodes a type III secretion system, and to a lesser extent the Shiga toxin, and causes outbreaks of gastroenteritis as well as sporadic infections [39]. In the wastewater samples, especially those collected from plant B, the abundance of *E. albertii* was extremely elevated after aerobic treatment (Figure 1), suggesting that *E. albertii* can survive in municipal wastewater. Hence, further studies are warranted to investigate the release of *E. albertii* from wastewater plants into aquatic environments. 

### 3.2. Relationship between Phylogroups and Antibiotic Resistance of E. coli

The prevalence of antibiotic resistance was highest in phylogroup B2. Furthermore, the prevalence of antibiotic resistance increased with time from 8% on day 0 to 39% on day 14 among the isolates in the wastewater samples collected from plant A (Figure 3). Similarly, in the wastewater sample collected from plant B antibiotic resistance was highest among the isolates classified as phylogroup B2 (Figure 3). In the plant B samples, although there was no increasing trend in the resistance rate, isolates classified as phylogroup B2 were the most prevalent on day 14 under aerobic conditions. Further evaluation of the relationship between phylogroups and antibiotic resistance found that *E*. *coli* isolates classified as phylogroup B2 were most capable to survive under aerobic conditions for 14 days. Previous studies have reported that the members of phylogroup B2 are more pathogenic than the members of the other phylogroups [40,41]. Based on the gene structure and sequencing data, the *E. coli* strains of phylogroup B2 differ from other strains and are more likely to cause parenteral infections [40,42,43]. Following the biological treatment of the municipal wastewater, although the abundance of *E. coli* had decreased by about 2log, the prevalence of antibiotic resistance increased due to the survival of the members of phylogroup B2, which were the most resistant to the tested antibiotics. 

The high survival rate of the strain in phylogroup B2 is common to the samples from plants A and B and has been suggested in previous studies [40]. The phylogroup B2 is an important example of *E. coli* phylogroups that survive in treated wastewater. On the other hand, the time-series changes in the prevalence of antibiotic-resistant and ESBL *E. coli* were completely different between the plant A and plant B samples. In the plant A sample, the prevalence of antibiotic-resistant *E. coli* increased by mixing the time (0–14 days) (Figure 2). In contrast, the prevalence of antibiotic-resistant *E. coli* increased at 7 days and then decreased sharply at 14 days in the plant B sample. These tendencies were also observed in the changes in the prevalence of ESBL *E. coli* (Figure 3). Although the cause of this difference has not been clarified, the fact that there is some kind of parameter unevenly affecting both sampling sites may be inferred. The changes in the *E. coli* and *Enterococci* counts during the aerobic treatment process differed significantly between the plant A and the plant B samples. In the sample from plant A the decrease between the 7th day and the 14th day was gradual and tended to a certain tendency, while in the sample from plant B the counts decreased linearly from day 0 to day 14. This indicates that the constituents of the substances and microorganisms in the two wastewater plants differed and progressed through different aerobic processes. It is possible that these differences also affected changes in the prevalence of antibiotic-resistant *E. coli*. In order to mention whether the prevalence increases in the municipal wastewater under aerobic conditions, it is necessary to consider adding other plant samples based on further results.

The phylogroups of the isolates from plant B were more complex and diverse compared with those from plant A. The survival rate of the phylogroup B2 isolates carrying beta-lactamase genes was relatively high under aerobic conditions. A previous study reported a significant increase in the abundance of CTX-M-producing *E. coli*, which is also a potential pandemic genotype [44]. The presence of clonal strains from the patient isolates of phylogenetic group B2 that can produce CTX-M may be responsible for the spread of ESBL resistance [45]. As a collection and storage site for pathogenic bacteria and pathogenic genes, wastewater plants provide an invaluable venue for the growth and spread of antibiotic-resistant bacteria, which is undoubtably increased by the constant agitation and flow of wastewater. Although bacteria gradually weaken and die during the wastewater treatment process, antibiotic-resistance genes can be acquired by bacteria with stronger survival capabilities. Municipal wastewater treatment plants can promote the spread of multiple antibiotic-resistant bacteria that are potentially harmful to human health and the environment. Hence, continued monitoring of antibiotic-resistant bacteria in wastewater treatment plants and sewage systems is needed. 

## 4. Materials and Methods

### 4.1. Sampling

Wastewater samples were collected from the sewage treatment plants A and B. Plant A treats sewage from a population of approximately 10,000 people with an average daily flow volume of 6600 m^3^. Plant B treats sewage from a population of approximately 163,000 people with an average daily flow volume of 92,500 m^3^. Plant A treats by rotary disc and oxidation ditch methods. Plant B treats by standard activated sludge methods. Samples were collected from plants A and B on 24 January 2018, and 15 January 2019, respectively, stored in sterile 5-L polyethylene bottles, and immediately transported for microbial and water quality analyses, which were conducted within 4 h.

The pH value, the electrical conductivity, and the turbidity were determined using a pH meter (HM-30G, DKK-TOA Corporation, Tokyo, Japan), a conductivity meter (CM30S, DKK-TOA Corporation, Tokyo, Japan), and a turbidity meter (SEP-PT-706D, Mitsubishi Chemical Corporation, Tokyo, Japan), respectively. In addition, the concentrations of total organic carbon (TOC) in the samples were determined by the state of the aerobic decomposition of organic substances using a TOC analyzer (TOC analyzer wet oxidation/non-dispersive infrared method model, Shimadzu Corporation, Kyoto, Japan).

### 4.2. Batch Mixing Experiment and Isolation of E. coli

Under batch aerobic conditions (in the dark at 20 °C), the wastewater was stirred and mixed (200 rpm) for two weeks and samples were collected on days 0, 7, and 14. In case of high concentrations of *E. coli* in the inflowing water the samples were diluted with sterilized physiological saline. All samples were filtered through a sterile cellulose ester membrane (pore, 0.45 µm; diameter, 47 mm; Toyo Roshi Kaisha, Ltd., Tokyo, Japan) and incubated on CHROMagar™ ECC plates (CHROMagar, Paris, France) at 37 °C for 24 h. Afterward, the mean numbers of colony-forming units (CFUs) of coliform bacteria and *E. coli* in each sample were determined from three replicates. Bacterial counts were expressed as CFU/100 mL. On the CHROMagar™ ECC plates the colonies of *E*. *coli* and coliform bacteria are blue and purple, respectively. One hundred *E. coli* isolates were collected from each sample on days 0, 7, and 14, thus 300 samples were collected from plant A and 300 from plant B. To confirm, all *E. coli* isolates were streaked on brain heart infusion (BHI) agar plates (1.5% agar, Becton, Dickinson and Company, Franklin Lakes, NJ, USA) and incubated at 37 °C for 24 h.

### 4.3. Identification of E. coli by Matrix-Assisted Laser Desorption/Ionization Coupled to Time-of-Flight Mass Spectrometry (MALDI-TOF MS)

The *E. coli*-positive isolates were pre-incubated in a nutrient liquid BHI agar medium at 37 °C for 18 h. The species were identified by MALDI-TOF MS [46]. An aliquot (1.0 mL) of the template was spotted directly into the wells of a 384-well stainless-steel target plate (MTP 384, Bruker Daltonics, Billerica, MA, USA), air-dried for 10 min, and then overlaid with 1.0 mL of matrix solution. All samples were analyzed using an Autoflex^®^ III TOF/TOF system (Bruker Daltonics, Billerica, MA, USA) operated in the linear positive mode within a mass range of 2000–20,000 Da according to the manufacturer’s instructions. For database construction and validation, measurements were obtained in the auto-execute mode using flexControl 3.0 software (Bruker Daltonics, Billerica, MA, USA) with the following parameters: linear positive, 3–20 kDa; detector gain, 1900 V; laser shots, 500–1000; and laser power, 30%. The instrument was calibrated using a Bruker bacterial test standard (part no. 8255343, Bruker Daltonics, Billerica, MA, USA).

Recorded mass spectra were processed with the MALDI Biotyper Compass microbial identification system (Bruker Daltonics, Billerica, MA, USA) using the standard settings. The MALDI Biotyper output is a log score value in the range of 0.000 to 3.000, representing the probability of correct identification of the isolate computed by a comparison of the peak list for an unknown isolate with the reference spectrum in the database. *E. coli* was identified by a log score value greater than 2.000.

### 4.4. Classification of Phylogroups of E. coli by Multiplex PCR

DNA samples were extracted from the *E. coli* isolates using the DNeasy Blood & Tissue Kit (QIAGEN, Valencia, CA, USA). The phylogroups (A, B1, B2, and D) were determined according to the multiplex PCR method reported by Clermont et al., [33], which is widely adopted as a simple but reliable *E. coli* phylotyping method [47].

The PCR reaction was conducted using the KAPA Taq extra PCR kit (KAPA Biosystems), primers for detection of three genes (*arpA*, *chuA*, and *yjaA*), and one DNA fragment (TspE4.C2), as presented in Appendix A. Each 20-µL reaction consisted of 4.3 µL of sterilized distilled water, 4.0 µL of 5 × KAPA Extra Buffer, 0.1 µL of Taq polymerase, 0.4 µL of deoxynucleoside triphosphates (dNTPs), 1 µL of each primer, and 2 µL of the DNA template. The concentration of our primers was 10 μM per primer, with a final primer concentration of 0.5 μM in the 20 μL total volume of PCR reaction. The PCR reaction was performed using a SimpliAmp™ Thermal Cycler (Thermo Fisher Scientific, Waltham, MA, USA) with the following reaction conditions: denaturation at 94 °C for 4 min, followed by 30 cycles at 94 °C for 5 s and 59 °C for 20 s, and a final extension step at 72 °C for 5 min. After the PCR reaction, 5 μL of the PCR product and 1 μL of 6 × loading buffer (Takara Bio, Inc., Shiga, Japan) were mixed, loaded into the wells of 2% agarose gels, and separated in 1× TBE buffer (Tris-Borate-EDTA buffer) with the Mupid^®^-One electrophoresis system (Nippon Genetics Co., Ltd., Tokyo, Japan) at 100 V for 40 min. After electrophoresis, the agarose gel was stained with ethidium bromide solution (0.05 µL/mL) for 10 min and then shaken in distilled water for 10 min to confirm the PCR amplification products. This method is based on the classification of amplification patterns of four primers (*arpA*, *chuA*, *yjaA*, and TspE4.C2) into each phylogroup. *E. coli* strain ATCC 25922 was used as a positive control for the PCR reaction.

### 4.5. Antimicrobial Susceptibility Testing

The minimum inhibitory concentration (MIC) of each antimicrobial agent was determined on Mueller–Hinton agar using the agar dilution method in accordance with the Clinical and Laboratory Standards Institute (CLSI) guidelines [48]. The *E. coli* isolates were cultured at 37 °C for 18 h in Mueller–Hinton broth (Becton, Dickinson and Company, Sparks, MD, USA) and then diluted to a final concentration corresponding to the 0.5 McFarland turbidity standard with fresh Mueller–Hinton broth. The *E*. *coli* isolates were then inoculated on the surface of 1.5% Mueller–Hinton agar containing graded concentrations of each antimicrobial in the wells of a microplate (Sakuma Co., Tokyo, Japan). Following incubation of the plates at 37 °C for 18 h, the MICs were determined. The MIC breakpoints for resistance were based on the CLSI criteria.

The antimicrobials used in the current study included ampicillin (AMP; graded concentrations of 4–64 µg/mL), gentamicin (GEN; 2–32 µg/mL), cefazolin (CFZ; 1–16 µg/mL), cefotaxime (CTX; 0.5–8 µg/mL), ceftazidime (CAZ; 2–32 µg/mL), tetracycline (TET; 2–32 µg/mL), imipenem (IMP), ciprofloxacin (CIP; 0.5–8 µg/mL), cefepime (CPM; 4–64 µg/mL) (Wako Pure Chemical Industries, Ltd., Osaka, Japan), and chloramphenicol (CHL; 4–64 µg/mL) (Sigma-Aldrich Corporation, St. Louis, MO, USA). Each of the tested agents was dissolved in distilled water or other appropriate solvents in accordance with the CLSI recommendations. The *E*. *coli* reference strain ATCC 25922 was used for quality control.

### 4.6. ESBL Genotypes of E. coli by Multiplex PCR

DNA was extracted by the same method as described above. The ESBL genotypes (TEM, SHV, CTX-M-1, CTX-M-2, and CTX-M-9) of the *E. coli* strains were determined by multiplex PCR as described elsewhere [49,50].

The PCR reaction was conducted with TaKaRa Taq^TM^ Hot Start Version (Takara Bio, Inc.), which consists of Takara LA Taq polymerase plus a monoclonal antibody, and the TEM, SHV, CTX-M-1, CTX-M-2, and CTX-M-9 primers (Appendix A). Each 50-µL reaction consisted of 29.75 µL of sterilized distilled water, 0.25 µL of TaKaRa Taq^TM^ Hot Start polymerase, 5.0 µL of 10 × PCR buffer (Mg^2+^ plus), 4.0 µL of dNTPs, 1.0 µL of each primer, and 1 µL of the DNA template. The concentration of our primers was 10 μM per primer, with a final primer concentration of 0.2 μM in the 50 μL total volume of PCR reaction. The PCR reaction was performed using a SimpliAmp™ Thermal Cycler (Thermo Fisher Scientific) under the following reaction conditions: denaturation at 94 °C for 2 min, followed by 30 cycles at 94 °C for 1 min, 55 °C for 1 min, 72 °C for 1.5 min, and a final extension step at 72 °C for 5 min. The PCR products were separated in 1× TBE buffer by agarose gel electrophoresis as described above.

## 5. Conclusions

The batch-type agitation and mixing experiments were conducted to monitor changes to *E. coli* strains and antibiotic-resistant profiles in wastewater samples under aerobic conditions. The survival rate of *E. coli* in phylogroup B2, accounting for approximately 73% of the isolates, was the highest in the aerobically treated wastewater in plant A after 14 days. In addition, B2 was the dominant phylogroup of antibiotic-resistant *E*. coli isolates at the end of the 14-day experimental period. Under aerobic conditions, the survival of the bacteria in the wastewater samples significantly differed among the strains and the prevalence of antibiotic-resistant *E. coli* had increased. Hence, the behavior of the members of phylogroup B2, which is the major phylogroup of *E. coli*, should be closely monitored.

## Figures and Tables

**Figure 1 antibiotics-11-00202-f001:**
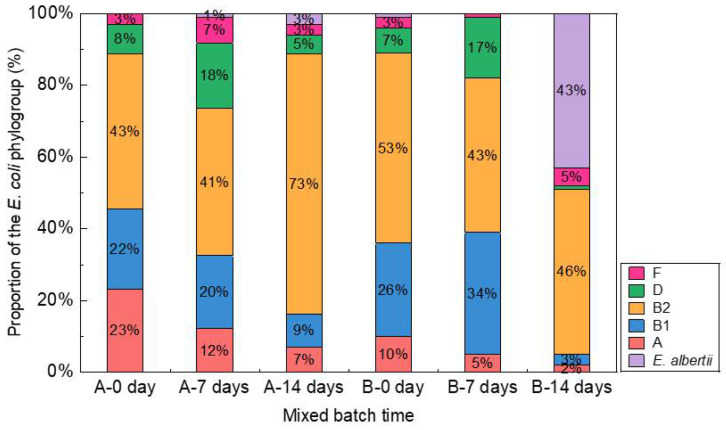
A bar chart of the cumulative percentage of *E. coli* isolates in wastewater samples collected from plants A and B in phylogenetic groups A, B1, B2, and D.

**Figure 2 antibiotics-11-00202-f002:**
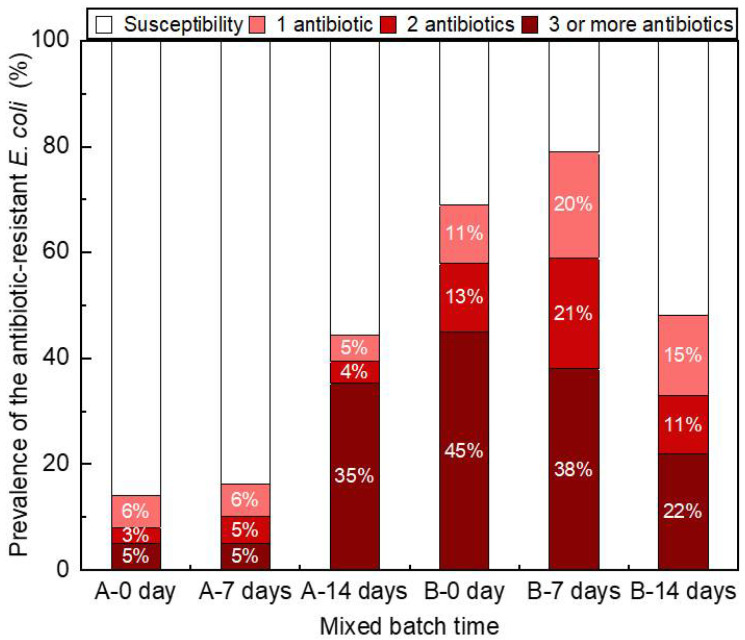
A bar chart of the cumulative percentage of *E*. *coli* isolates from plants A and B that were resistant to 1, 2, and 3 or more antibiotics.

**Figure 3 antibiotics-11-00202-f003:**
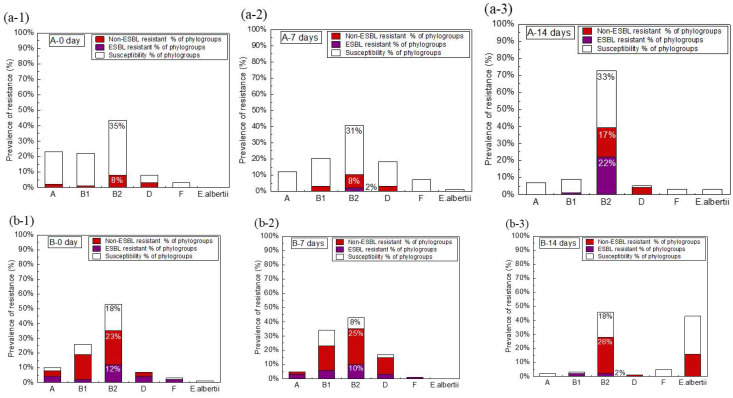
The relationship between phylogroups and antibiotic resistance (ESBL and non-ESBL) of *E. coli* isolates in wastewater samples collected from plants A and B. *E. coli* isolates obtained from plant A samples on days 0 (**a-1**), 7 (**a-2**), and 14 (**a-3**). *E. coli* isolates obtained from plant B on days 0 (**b-1**), 7 (**b-2**), and 14 (**b-3**).

**Table 1 antibiotics-11-00202-t001:** The detection of phylogroups and ESBL-producing *E. coli* isolates.

Isolates from Plant A and B	ESBL Types by the Following PCR Type					Phylogroup
	TEM	SHV	CTX-M-1	CTX-M-2	CTX-M-9	
A-7 days-91	N	N	N	N	Y	B2
A-7 days-100	N	N	N	Y	Y	B2
A-14 days-5	N	N	Y	N	N	B2
A-14 days-7	N	N	N	Y	N	B2
A-14 days-8	N	N	N	N	Y	B2
A-14 days-9	N	N	N	N	Y	B2
A-14 days-21	N	N	N	Y	Y	B2
A-14 days-23	N	N	N	N	Y	B2
A-14 days-42	N	N	N	N	Y	B2
A-14 days-54	N	N	N	N	Y	B2
A-14 days-59	N	N	N	N	Y	B2
A-14 days-62	N	N	N	N	Y	B2
A-14 days-71	N	N	N	N	Y	B2
A-14 days-73	N	N	N	N	Y	B2
A-14 days-80	N	Y	N	N	N	B2
A-14 days-82	N	Y	N	N	N	B2
A-14 days-83	N	N	Y	Y	N	B2
A-14 days-84	N	N	Y	Y	N	B2
A-14 days-85	N	N	Y	N	N	B2
A-14 days-86	N	N	N	N	Y	B2
A-14 days-87	N	N	N	N	Y	B2
A-14 days-88	N	N	N	N	Y	B2
A-14 days-92	N	N	N	N	Y	B2
A-14 days-94	N	N	N	N	Y	B2
A-14 days-100	N	Y	N	N	N	B1
B-0 day-1	N	N	N	N	Y	B2
B-0 day-2	N	N	N	N	Y	D
B-0 day-3	N	N	N	Y	N	B2
B-0 day-4	N	Y	N	N	N	B2
B-0 day-12	N	N	N	N	Y	D
B-0 day-13	N	Y	N	N	N	D
B-0 day-14	N	N	N	N	Y	F
B-0 day-23	N	N	N	N	Y	B2
B-0 day-24	N	N	Y	N	N	B2
B-0 day-25	N	N	N	N	Y	B2
B-0 day-26	N	Y	N	N	N	A
B-0 day-38	N	N	N	Y	Y	B2
B-0 day-39	N	Y	N	N	N	B1
B-0 day-43	N	Y	N	N	N	A
B-0 day-60	N	N	N	Y	Y	B2
B-0 day-62	N	Y	N	N	N	B2
B-0 day-64	N	Y	N	N	N	B2
B-0 day-65	N	Y	N	N	N	A
B-0 day-66	N	Y	N	N	N	A
B-0 day-85	N	N	N	N	Y	D
B-0 day-86	N	Y	N	N	N	B1
B-0 day-87	N	N	N	N	Y	B2
B-0 day-88	N	Y	N	N	N	F
B-0 day-89	N	Y	N	N	N	B2
B-7 days-17	N	N	N	Y	N	B2
B-7 days-18	N	Y	N	N	N	B1
B-7 days-19	N	N	Y	N	Y	B1
B-7 days-20	N	Y	N	N	Y	F
B-7 days-26	N	Y	N	N	Y	D
B-7 days-40	N	Y	Y	N	N	B2
B-7 days-45	N	Y	N	N	Y	B2
B-7 days-46	N	Y	N	N	Y	B2
B-7 days-47	Y	Y	N	N	N	A
B-7 days-48	N	Y	N	N	N	D
B-7 days-49	N	N	N	Y	N	B2
B-7 days-50	N	N	N	N	Y	A
B-7 days-51	N	Y	Y	N	N	A
B-7 days-53	N	Y	N	N	N	B1
B-7 days-58	N	N	N	N	Y	D
B-7 days-67	N	N	N	N	Y	B2
B-7 days-68	N	N	N	N	Y	B2
B-7 days-69	N	N	N	Y	Y	B1
B-7 days-70	N	Y	Y	N	N	B1
B-7 days-71	N	N	N	N	Y	B1
B-7 days-84	N	Y	N	N	N	B2
B-7 days-88	N	Y	N	N	N	B2
B-7 days-91	Y	Y	N	N	Y	B2
B-14 days-27	N	Y	N	N	N	B2
B-14 days-28	N	N	N	Y	N	B1
B-14 days-30	N	Y	N	N	N	B1
B-14 days-37	N	N	N	Y	Y	B2

Y (yes) or N (no) to indicate whether it is an ESBL genotype.

## Data Availability

All data generated or used during the study appear in the submitted article or Appendix A.

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
