# Peer review of "Persistence of Antibiotic-Resistant Escherichia coli Strains Belonging to the B2 Phylogroup in Municipal Wastewater under Aerobic Conditions"

_antibiotics, 2022, doi:10.3390/antibiotics11020202_

Round 1

Reviewer 1 Report

The authors describe the analysis and characterization of E. coli present in wastewater samples, and the association of phylogroups with antimicrobial resistance. Characterization of microbial pathogens, particularly in regards to their antimicrobial resistance, is of high interest. Some issues must be addressed prior for being acceptable for publication.

Major issues:

Even though the authors have analyzed a relatively large number of isolates, it must be kept in mind that only 2 settings were included in the study. In line with this, even though it is correct that phylogroup B2 is the most predominant, the percentage seemed to have significantly increased in setting A while decreasing in setting B after 14 days, thus having an additional sampling site would be helpful to better see if there is a particular trend. The fact that there is some kind of parameter affecting unevenly both sampling sites may be inferred from the author's Figure S1-b1 and S1-b2, as while in site A the decrease in E. coli and Enterococci is "slight" between day 7 and 14 it is very abrupt in site B. Considering all this, having an additional site would help strengthening the conclusions. 

Minor issues:

Lines 35-37. Rewrite for clarity.

Line 60. Great??

Line 119. Following

Line 138. and throughout the manuscript. For PCR reactions provide primer concentration, not volume.

Line 145. Which buffer was used for GE?

Line 205. What are "pseudopositive strains"?

Line 226. Phylogroup.

General comments:

Revise species/ genus names formats throughout the manuscript. Keep in mind they must be written in italics (e.g. lines 39 and 42).

Author Response

Thank you for the review and comments. This major issue is a critical point for this study. Due to time and other factors, this study did only compare the conclusions of the two wastewater treatment plants. As you mentioned, adding having an additional survey site would indeed be very convincing for the conclusions. The conclusion of this study is mainly to show that under aerobic conditions, the resistant phylogroup B2 remained the largest part of the major phylogroups (A, B1, B2, D, F) of E. coli after two weeks. However, as you pointed out, the changes and trends in overall and ESBL prevalence percentages of E. coli differed significantly in the A and B samples. The reason for this had be unclear. However, there was a hint in your comment. Therefore, we have added a consideration about the causes of different prevalence in the A and B samples in the section 3. 2 and revised slightly in Abstract.

[Revised MS Line 243–262]

High survival of strain phylogroup B2 is common to the samples of plant A and plant B and has been suggested in previous studies (40). The phylogroup B2 is important as E. coli phylogroups that survive in treated wastewater. On the other hand, the time-series changes in the prevalence of antibiotic-resistant and ESBL E. coli were completely different between the plant A and plant B samples. In the plant A sample, the prevalence of antibiotic-resistant E. coli increased by mixing time (0–14 days) (Figure 2). In contrast, the prevalence of antibiotic-resistant E. coli increased at 7 days and then decreased sharply at 14 days in the plant B sample. These tendencies were also observed in changes in the prevalence of ESBL E. coli (Figure 3). Although the cause of this difference has not been clarified, the fact that there is some kind of parameter affecting unevenly both sampling sites may be inferred. The changes in E. coli and Enterococci counts during the aerobic treatment process differed significantly between the plant A and plant B samples. In the sample of plant A, the decrease on the 7 days to 14th days became gradually and tended to a certain tendency, while in the sample of plant B, the counts decreased linearly from the 0 day to 14 days. This indicates that the constituents of the substances and microorganisms in two wastewater plants differed and would progress through different aerobic processes. It is possible that these differences also affected changes in the prevalence of antibiotic-resistant E. coli. In order to mention whether the prevalence increases in municipal wastewater under aerobic conditions, it is necessary to consider based on the further results adding other plant samples.

Minor issues:

Lines 35-37. Rewrite for clarity.

=> The sentences have been revised as follow. [Revised MS Line 21–25]

After batch-mixing experiments under aerobic conditions, the surviving antibiotic-resistant E. coli included mainly of multidrug-resistant and beta-lactamase-producing isolates belonging to phylogroup B2. These results suggest that phylogroup B2 isolates that have acquired antibiotic resistance had a high survivability in treated wastewater.

Line 60. Great??

=> Sorry. We have corrected.

Line 119. Following

=> Sorry. We have corrected.

Line 138. and throughout the manuscript. For PCR reactions provide primer concentration, not volume.

=> The primer concentrations for PCR reactions have been added.

[Revised MS Line 340]

The concentration of our primers was 10 μM per primer, with a final primer concentration of 0.5 μM in the 20 μL total volume of PCR reaction.

[Revised MS Line 382]

The concentration of our primers was 10 μM per primer, with a final primer concentration of 0.2 μM in the 20 μL total volume of PCR reaction.

Line 145. Which buffer was used for GE?

=> The buffer for GE has been added.

[Revised MS Line 347, Line 387]

1x TBE buffer (Tris-Borate-EDTA buffer) was used for gel electrophoresis.

Line 205. What are "pseudo positive strains"?

=> Pseudo positive strains means false-positive strain. In this study, E. coli isolates were collected from each sample of wastewater using CHROMagar™ ECC selection medium. This medium has a 97% sensitivity to E. coli isolates (Ogden et al. 1991). To avoid false positive errors during counting and isolation, we further used MALDI-TOF MS for identification at the molecular level of the bacteria to determine the positive E. coli.

Line 226. Phylogroup.

=> We have corrected.

General comments:

Revise species/ genus names formats throughout the manuscript. Keep in mind they must be written in italics (e.g. lines 39 and 42).

=> We have checked and revised as your suggestion.

Reviewer 2 Report

The paper overall is written in a very mismanaged way, The article didn't follow the journal instructions at all. I don't know how the article was passed from initial screening to sent for review. It should be directly rejected on the initial screening. First of all the whole paper cited the references with the way ABC et al while the MDPI references style is [1] and at the end, the bibliography is mentioned with [1],[2] . . . . . 

The whole paper needs to be arranged from start till the end and then should be resubmitted for review again. No wonder the article contains good data to be considered but it should be reconsidered after revision.

Further comments are mentioned below.

  1. After the introduction section. It should be results and then discussion and then Materials and methods. Kindly follow the instruction of the authors.
  2. The fig.1 and Fig.2 are very blurred kindly redraw it. 
  3. In supplementary data after Table 3. I don't know what is there is it a figure or table? no caption and the digits are very small I cannot see anything clear.

Author Response

We are sorry that we submitted the paper without according to the MDPI format of the submission rules. We hurried because the posting rules weren't mandatory. With the support of the MDPI editors, the paper has been revised to match the MDPI regulations. The item you pointed out has also been corrected.

Further comments are mentioned below.

  1. After the introduction section. It should be results and then discussion and then Materials and methods. Kindly follow the instruction of the authors.

=> The structure of the paper has been revised according to your suggestions.

  1. The fig.1 and Fig.2 are very blurred kindly redraw it.

=> The figures have been shown clearly.

  1. In supplementary data after Table 3. I don't know what is there is it a figure or table? no caption and the digits are very small I cannot see anything clear.

=> We have added the caption for each Figure and Table in Supplementary data.

Reviewer 3 Report

In the manuscript titled "Persistence of antibiotic-resistant Escherichia coli strains belonging to the B2 phylogroup in municipal wastewater under aerobic conditions", the authors raise a very interesting issue. E.coli phylogroup B2 are known for their widespread pathogenicity and it is important to monitor and survey their prevalence and dissemination in the environment. More importantly, the authors find that these strains persist even after wastewater treatment.

This has a major implication on public health and subsequent necessity for modifying existing wastewater treatment methods.

The authors tackle this with a sound methodology and analysis and the manuscript is very well written.

Author Response

Thank you for the review and the kind comment. We would like to express our sincere gratitude for your understanding of the purpose and significance of this paper.

Round 2

Reviewer 2 Report

The authors addressed all the questions clearly and they improved a paper a lot. The article is scientifically strong and i recommend for publication.